# Considerations for Developing a Reassessment Process: Report from the Canadian Real-World Evidence for Value of Cancer Drugs (CanREValue) Collaboration's Reassessment and Uptake Working Group

Wei Fang Dai [1,2,†], Vanessa Arciero [1,†], Erica Craig [3], Brent Fraser [4], Jessica Arias [5], Darryl Boehm [6], Nevzeta Bosnic [7], Patricia Caetano [8], Carole Chambers [9], Barry Jones [10], Elena Lungu [7], Gunita Mitera [11], Tanya Potashnik [7], Anthony Reiman [12,13,14], Trevor Ritcher [4], Jaclyn M. Beca [2,5], Avram Denburg [15], Rebecca E. Mercer [2,5], Ambica Parmar [16], Mina Tadrous [17], Pam Takhar [5], Kelvin K. W. Chan [1,2,16,*] and on behalf of the CanREValue Collaboration Reassessment and Uptake Working Group [‡]



[1] Temerty Faculty of Medicine, University of Toronto, 1 King's College Circle, Toronto, ON M5S 1A8, Canada; weifang.dai@mail.utoronto.ca (W.F.D.); vanessa.arciero@mail.utoronto.ca (V.A.)
[2] Canadian Centre for Applied Research in Cancer Control, Toronto, ON M5G 2L3, Canada; Jaclyn.Beca@ontariohealth.ca (J.M.B.); rebecca.mercer@ontariohealth.ca (R.E.M.)
[3] New Brunswick Cancer Network, Saint John, NB E2J 3S4, Canada; Erica.Craig@gnb.ca
[4] Canadian Agency for Drugs and Technologies in Health, Ottawa, ON K1S 5S8, Canada; BrentF@cadth.ca (B.F.); TrevorR@cadth.ca (T.R.)
[5] Ontario Health (CCO), Toronto, ON M5G 2L7, Canada; jessica.arias@ontariohealth.ca (J.A.); pam.takhar@ontariohealth.ca (P.T.)
[6] Saskatchewan Cancer Agency, Regina, SK S4W 0G3, Canada; Darryl.Boehm@saskcancer.ca
[7] Patented Medicine Prices Review Board, Ottawa, ON K1P 1C1, Canada; nevzeta.bosnic@pmprb-cepmb.gc.ca (N.B.); elena.lungu@pmprb-cepmb.gc.ca (E.L.); tanya.potashnik@pmprb-cepmb.gc.ca (T.P.)
[8] Government of Manitoba, Winnipeg, MB R3B 3M9, Canada; Patricia.Caetano@gov.mb.ca
[9] Alberta Health Services, Edmonton, AB T5J 3E4, Canada; Carole.Chambers@albertahealthservices.ca
[10] Health Canada, Ottawa, ON K1Y 4X2, Canada; barry.jones@canada.ca
[11] Canadian Association of Provincial Cancer Agencies, Toronto, ON M5H 1J9, Canada; gmitera@capca.ca
[12] Department of Medicine, Dalhousie University, Halifax, NS B3H 2Y9, Canada; Anthony.Reiman@HorizonNB.ca
[13] Department of Biology, University of New Brunswick, Fredericton, NB E3B 5A3, Canada
[14] Department of Oncology, Saint John Regional Hospital, Saint John, NB E2L 42L, Canada
[15] The Hospital for Sick Children, Toronto, ON M5G 1X8, Canada; avram.denburg@sickkids.ca
[16] Sunnybrook Health Sciences Centre, Toronto, ON M4N 3M5, Canada; ambika.parmar@sunnybrook.ca
[17] Women's College Hospital, Toronto, ON M5S 1B2, Canada; Mina.Tadrous@wchospital.ca
[*] Correspondence: kelvin.chan@sunnybrook.ca
[†] Wei Fang Dai and Vanessa Arciero contributed equally to this work.
[‡] Membership of the CanREValue Collaboration Reassessment and Uptake Working Group is provided in the Acknowledgments.

**Abstract:** The Canadian Real-world Evidence for Value in Cancer Drugs (CanREValue) Collaboration was established to develop a framework for generating and using real-world evidence (RWE) to inform the reassessment of cancer drugs following initial health technology assessment (HTA). The Reassessment and Uptake Working Group (RWG) is one of the five established CanREValue Working Groups. The RWG aims to develop considerations for incorporating RWE for HTA reassessment and strategies for using RWE to reassess drug funding decisions. Between February 2018 and December 2019, the RWG attended four teleconferences (with follow-up surveys) and two in-person meetings to discuss recommendations for the development of a reassessment process and potential barriers and facilitators. Modified Delphi methods were used to gather input. A draft report of recommendations (to December 2018) was shared for public consultation (December 2019 to January 2020). Initial considerations for developing a reassessment process were proposed. Specifically, reassessment can be initiated by diverse stakeholders, including decision makers from public drug plans or industry

stakeholders. The reassessment process should be modelled after existing deliberation and recommendation frameworks used by HTA agencies. Proposed reassessment outcome categories include maintaining status quo, revisiting funding criteria, renegotiating price, or disinvesting. Overall, these initial considerations will serve as the basis for future advancements by the Collaboration.

**Keywords:** health technology assessment; real-world evidence; reassessment

## 1. Introduction

Health technology assessment (HTA) has traditionally been viewed as a process for evaluating new technologies prior to funding [1]. Recently, an international joint task group, the International Network of Agencies for Health Technology Assessment and Health Technology Assessment International, made recommendations to refine and broaden the definition of HTA to include the management of a health product throughout its lifecycle, from pre- to post-funding [2]. A fundamental element of lifecycle HTA is reassessment, defined as "a structured, evidence-based assessment of the clinical, social, ethical, and economic effects of a technology currently used in the healthcare system, to inform optimal use of that technology in comparison to its alternatives" [3].

Real-world evidence (RWE), derived from real-world data [4], has been suggested by stakeholders as being potentially valuable for facilitating systematic evidence-based reassessment to enable lifecycle HTA and improve reimbursement decision making [5–9]. An inherent strength of RWE is the unselected patient population that may be more relevant to routine practice, as randomized clinical trials (RCTs) often have selective inclusion criteria, which contributes to the variation between efficacy observed in trials and effectiveness observed in the real world [4,10–15]. Since initial drug funding decisions are based on clinical benefit observed from RCTs and the value of a drug is estimated using economic models, a large efficacy–effectiveness gap is particularly important to decision makers [16]. RWE can provide insight into the true value of drugs in actual use, which can serve to inform funding decisions, especially for the purpose of reassessment.

The Canadian Real-world Evidence for Value in Cancer (CanREValue) Collaboration was established to develop a framework for generating and applying RWE to inform cancer drug funding decisions [17]. One goal of the framework is to enable the reassessment and refinement of funding decisions that may inform renegotiation or disinvestment by decision makers across Canada [17]. The CanREValue Reassessment and Uptake Working Group (RWG), one of five formal working groups established by the CanREValue Collaboration, aims to develop a process for incorporating RWE into HTA reassessment and for providing advice on strategies for incorporating RWE into policy decisions [17]. The mandate of RWG is to develop a conceptual process for incorporating the reassessment of funded cancer drugs into the current Canadian healthcare system and for recommending the processes and factors required for revisiting price negotiations and funding decisions [17]. This paper describes the issues considered by the CanREValue RWG in its work to develop a preliminary reassessment process based on RWE.

## 2. Materials

### 2.1. Working Group Formation

The RWG is composed of a diverse group of experts with experience in cancer drug funding and an interest in RWE. Members include clinicians, patients and caregivers, researchers, decision makers (provincial Ministry of Health and cancer agency), HTA agencies (CADTH, Institut national d'excellence en santé et en services sociaux (INESSS)), regulatory bodies (Health Canada, the Patented Medicine Pricing Review Board), the pan-Canadian Pharmaceutical Alliance, and the Canadian Association of Provincial Cancer Agencies.

## 2.2. Approach to Framework Development

Modified Delphi methods [18] were used to guide discussion and gather collective opinion amongst the RWG. Four teleconferences and two in-person meetings were held between February 2018 and December 2019 (Table 1). Following each teleconference, the RWG were invited to complete a survey developed by the RWG Chairs and core CanREValue Collaboration research team based on main discussion points from the teleconferences (Supplementary Materials). Each survey was available for two weeks, and at least one reminder was sent to the RWG. Responses were collated, and major themes summarized.

**Table 1.** Timeline and Specific Aims of Reassessment and Uptake Working Group (RWG) Meetings and Surveys.

| Meeting | Aim(s) |
|---------|--------|
| Round 1: February 2018<br>First teleconference and post-meeting survey | Introduce RWG to the purpose of the framework and identify the main components of a framework for reassessment |
| Round 2: April 2018<br>Second teleconference and post-meeting survey | Identify areas of consensus surrounding the reassessment process following an in-depth discussion of a draft framework for reassessment, collated from Round 1 survey responses |
| In-Person Meeting: May 2018 | Joint meeting between the Policy WGs * to discuss areas of agreement and discrepancy following rounds 1 and 2, and discuss progress to date to ensure alignment in the development of the framework |
| Round 3: September 2018<br>Third teleconference and post-meeting survey | Discuss how a reassessment could be operationalized |
| In-Person Meeting: May 2019 | Joint meeting between the Policy WGs * to host a mock reassessment |
| Round 4: December 2019<br>Fourth teleconference and post-meeting survey | Discuss barriers and facilitators to implementation of recommendations for reassessment |

* Policy WGs: CanREValue Collaboration RWG; CanREValue Collaboration Planning and Drug Selection WG.

## 2.3. External Stakeholder Consultation

Content developed by the RWG (to 2019) was compiled into an interim policy report for external stakeholder consultation. The report was distributed via the CanREValue Collaboration mailing list and website, and it was promoted through the official CanREValue Twitter account (@CanREValue). Specific consultation questions were provided to guide stakeholder feedback (Supplementary Materials S1), and respondents were encouraged to provide additional comments. Public feedback was collected via email between December 2019 and January 2020. Responses were collated, and main themes were summarized by the core CanREValue Collaboration research team and shared with the RWG. Responses to public feedback are available on the CanREValue website (https://cc-arcc.ca/wp-content/uploads/2020/10/CanREvalue-Interim-Policy-WG_Response-to-Stakeholder-Consultation_30Oct2020-FINAL.pdf, accessed on 28 August 2021).

## 3. Findings

### 3.1. Considerations from the Reassessment and Uptake Working Group

Through an iterative, consultative process, the RWG discussed and reached agreement on four main considerations when developing a reassessment process (Table 2).

**Table 2.** Summary of Four Main Considerations when Developing a Reassessment Process.

| Recommendation | Summary | External Stakeholder Feedback and Responses from the Reassessment and Uptake Working Group (RWG) |
|---|---|---|
| 1. The Process of Reassessment | Reassessment can be initiated by decision makers and industry<br>Reassessment should be conducted by HTA agency<br>Reassessments should undergo an eligibility review and prioritization | Feedback: Diverse perspectives should be considered throughout (such as, patients, decision makers, HTA agencies, clinicians, methodologists, and manufacturers)<br>Response: RWG agreed with this recommendation<br>Feedback: Current CADTH process for disseminating and sharing outcomes at each stage should be adopted<br>Response: RWG agreed with this recommendation |
| 2. Evaluation and Deliberation of Evidence for Reassessment | A model similar to the current CADTH reimbursement review expert committee deliberation and recommendation frameworks should be adopted<br>Regulators, academia, research organizations, and/or F/P/T jurisdictions should collaborate when deliberating the evidence<br>Evidence to consider includes gaps in initial drug funding recommendations, utilization trends and indication creep (i.e., use of drug beyond the originally recommended population), patient experience, clinical outcomes, real-world cost-effectiveness, changes in the funding algorithm and treatment sequencing, and operational factors (i.e., implementation and sustainability) | Feedback: Ethics should have a designated evidence category during the review process<br>Response: Ethical considerations and oversight should be embedded throughout the reassessment process |
| 3. Reassessment Outcome Categories | Reassessment outcomes were proposed to be summarized in three categories: (1) status quo (i.e., continue funding), (2) revisit funding criteria or pricing, and (3) do not continue funding/delist | Feedback: Removal of "do not continue funding/delist" category as it may threaten medication access<br>Response: Unlikely that this recommendation will be made in the absence of strong evidence |
| 4. Barriers and Facilitators to the Implementation of Recommendations for Reassessment | Barriers to implementation included evidence generation, clinical context barriers, system level barriers, and general barriers<br>Facilitators to implementation included generation of high-quality RWE, clearly defined reassessment criteria and outcome categories, collaboration, and general facilitators. | |

HTA: Health technology assessment; CADTH: Canadian Agency for Drugs and Technology and Health; F/P/T: Federal, provincial, territorial.

### 3.1.1. Consideration 1: The Process of Reassessment Review

Reassessment can be initiated by decision makers from cancer agencies and public drug plans and by industry stakeholders. Reassessment may be identified by federal, provincial, or territorial (F/P/T) public drug programs when uncertainties are identified during the initial drug reimbursement review. Industry stakeholders may also propose reassessment at the initial funding stage, or when new evidence about a funded drug

emerges. Given the rapidly evolving treatment landscape in some therapeutic space, a jurisdictional committee, such as one that includes public drug program representatives, may determine that an initial proposed reassessment is no longer relevant. This may occur following the introduction of a new comparator treatment that is unequivocally superior to the drug under consideration, or after a change in the funding algorithm that makes the drug under consideration for review no longer the standard of care (i.e., the drug under consideration has been replaced by another drug).

The RWG recommended that HTA agencies (CADTH or INESSS) should lead and conduct the reassessment. When HTA agencies conduct the initial review of a file, uncertainties surrounding the clinical use and economic value of a drug can be identified and a recommendation to collect future evidence to minimize uncertainty around clinical benefit or cost effectiveness can be suggested. For the purposes of reassessment, HTA agencies can also provide guidance regarding the evidence and data required for the reassessment and who should be responsible for collecting the data and initiating the review. Based on context and available information at the time of recommendation (e.g., disease and drug under consideration, treatment landscape), HTA agencies can also provide recommendations regarding a suitable timeline for reassessment. The circumstances under which a candidate drug can be considered for future data collection and subsequent reassessment include (1) a high unmet need (i.e., no reasonable, publicly funded treatment alternatives are available) or (2) substantial uncertainty in the magnitude of the clinical benefit of the drug that can be feasibly addressed by a RWE study.

When the reassessment process is initiated by a stakeholder, the file should undergo an eligibility review and prioritization. In the current HTA process, resubmissions to CADTH undergo an eligibility assessment to ensure that new information addresses key issues from the initial recommendation [16]. A similar eligibility assessment should be undertaken for reassessments. Drugs that have received an initial negative recommendation would not be eligible for the reassessment process since CADTH currently has a resubmission process to accommodate these files. Moreover, reassessment should not be conducted for all funded drugs. Consequently, a process for prioritization of reassessments would need to be established. Jurisdictional group(s), such as CADTH's Provincial Advisory Group [19], could inform prioritization of drugs for data collection and reassessments. The feasibility of developing a process for identifying and prioritizing potential uncertainties that can be addressed by real-world studies is within the mandate of the CanREValue Collaboration's Planning and Drug Selection WG [17].

Two reassessment streams—a standard "comprehensive review" and a more narrowly focused "tailored review"—were initially considered. It was concluded that at this stage, a single comprehensive review stream is a reasonable initial approach. Without experience, it may be difficult to identify instances that would warrant a tailored rather than a comprehensive reassessment. However, integrating a tailored review stream may be considered in the future.

Transparency is critical when conducting reassessment, and an engagement model similar to that used by HTA agencies during initial drug reviews—which includes patients, clinicians, decision makers, and industry—was considered for adoption, as it is important to reengage with all stakeholders involved in the initial review. Nonetheless, it is recognized that the expertise required to review new evidence during a reassessment may vary from that required in initial reviews. For example, initial reviews largely involve RCT evidence and economic models, whereas reassessments should consider any data that may resolve uncertainty. It will likely require further exploration to determine if similar procedures should be followed for reassessments.

3.1.2. Consideration 2: Evaluation and Deliberation of Evidence for Reassessment

When considering the type of evidence that should be examined during a reassessment, a model similar to the current deliberative framework employed by the CADTH expert committee for initial drug recommendation was recommended. At the initial drug

review, committee experts review clinical and economic evidence. They also deliberate on patient values in addition to ethical, implementational, and adoption-related considerations. It was emphasized that reassessments maintain the same high-quality evidence standards as the initial drug review. HTA agencies will need to collaborate with regulators, academia, research organizations, and/or F/P/T jurisdictions when deliberating the evidence. Some of the evidence that should be considered during reassessments includes gaps in the evidence that informed the initial drug funding recommendation, utilization trends and indication creep (i.e., use of drug beyond the originally recommended population), patient experience, clinical outcomes, real-world cost-effectiveness, changes in the funding algorithm and treatment sequencing, and operational factors (i.e., implementation and sustainability).

### 3.1.3. Consideration 3: Reassessment Outcome Categories

Three categories for reassessment outcomes were proposed: (1) status quo (i.e., continue funding), (2) revisit funding criteria or pricing, and (3) do not continue funding/delist (Table 3).

**Table 3.** Description of Reassessment Outcome Categories.

| Outcome Category | Description |
|---|---|
| Status Quo | • Data provided for the reassessment confirmed the effectiveness, safety, and cost-effectiveness of the initial review of the investigated drug, and thus there is no need to change the current reimbursement recommendation, or<br>• Data provided for the reassessment was insufficient to address an important question of effectiveness or cost-effectiveness, and additional data and subsequent reassessment is required |
| Revisit Funding Criteria or Pricing | • Data provided for the reassessment warranted a revision to the criteria for funding (i.e., broader or narrower indication), and/or<br>• Data provided for the reassessment modified the cost effectiveness of the drug (i.e., the drug performed better or worse than expected on one or more key outcomes of interest), and jurisdictions should evaluate whether existing pricing agreements need to be revised |
| Do Not Continue to Fund/Delist | • Data provided for the reassessment confirmed that there was at least one superior alternative treatment available, based on patient preference, effectiveness, safety, and/or cost-effectiveness |

### 3.1.4. Consideration 4: Barriers and Facilitators to the Implementation of Recommendations for Reassessment

Specific barriers and facilitators to the implementation of recommendations for reassessment were discussed (Table 4). Identified barriers included evidence generation, clinical context barriers, and system-level barriers. Similarly, facilitators to the implementation of recommendations for reassessment included generation of high-quality RWE, clearly defined reassessment criteria and outcomes, and collaboration.

**Table 4.** Barriers and Facilitators to the Implementation of Recommendations for Reassessment.

| Barriers to the Implementation of Recommendations for Reassessment | |
|---|---|
| Evidence Generation | • Unavailability, unreliability, and poor quality of RWE<br>• Low population size and treatment volume |
| Clinical Context | • Availability of alternative treatment options<br>• Optimal sequencing of treatments in the therapeutic space<br>• Changes in the treatment landscape (i.e., increasing number of biosimilars and generic drugs, treatment alternatives) |
| System Level | • Balancing and weighting the economic evidence relative to the clinical evidence<br>• Resistance from manufacturers and/or patients<br>• Lack of willingness of decision makers to delist a drug, or increase funding relative to benefit<br>• Timing of renegotiation of drug price alongside other negotiations<br>• Potential perceived conflict of interest that may result from the involvement of manufacturers in the RWE programs<br>• Varying funding criteria amongst provincial drug plans<br>• Establishing a threshold for narrowing the indication or delisting a drug |
| General | • Inadequate available resources<br>• Lack of standards, communication, and education on the topic of RWE amongst stakeholders<br>• Lack of clarity on the rationale for the original recommendation<br>• Lack of strategies for implementation |
| Facilitators of the Implementation of Recommendations for Reassessment | |
| Generation of High-Quality RWE | • Prospective data collection<br>• Data on utilities to facilitate a more robust model and support negotiation<br>• Partnerships with organizations to collect and analyze real-world data (e.g., utilization data) |
| Clearly Defined Reassessment Criteria and Outcomes | • Clear criteria for assessment outcomes developed by stakeholder consultation<br>• Consideration of "outcome ranges" when renegotiations may be desirable |
| Collaboration | • Agreed upon study protocols by manufacturers and decision makers<br>• Collaborative efforts with manufacturers to leverage patient access program data<br>• Engagement with the jurisdictions that the recommendations will be implemented |
| General | • Timeliness of study results to address specific questions<br>• Ability to communicate and educate about the benefit of using RWE |

RWE: Real-world evidence.

### 3.2. Feedback from External Stakeholder Consultation

During the external consultation process, 21 respondents provided feedback. All feedback was carefully considered by the RWG and summarized below and in Table 2.

It was proposed through the external stakeholder consultation process that ethics should have a designated evidence category during the review process. RWG recognizes the importance of ethical considerations in the context of the reassessment process and believes that ethical considerations and oversight should be embedded throughout the reassessment process, just as ethics, equity, and fairness are incorporated in all other HTA processes in Canada. Ethics is a core principle of both CADTH and INESSS, and the RWG expects that this standard would also apply to any new assessment processes [16].

Some respondents suggested the removal of the recommendation category "do not continue funding/delist", as it may threaten access to medication. Given the potential outcome categories for a reassessment, RWG acknowledged that it is unlikely that a suggestion will be made to discontinue funding or delist a drug in the absence of strong evidence against its use; specifically, if the drug does not provide added benefit or leads to net harm in comparison with other available treatment options. This recommendation would be made only after careful consideration of potential advantages and disadvantages of use for all patients.

It was suggested that the reassessment framework should adopt the current CADTH process for disseminating outcomes at each stage of reassessment. Feedback also emphasized the need for considering diverse perspectives throughout the development and implementation of a reassessment process, including patients, decision makers, HTA agencies, clinicians, methodologists, and manufacturers. RWG recognizes the value of this inclusive approach and have proactively established the CanREValue Engagement WG to ensure key stakeholders are engaged in the development and implementation of the framework [17].

## 4. Discussion

To our knowledge, the CanREValue Collaboration is the first to assemble the expertise and perspectives of diverse stakeholders to form the RWG with the aim of developing a process for the reassessment of funded cancer drugs in Canada. We outlined the main considerations, barriers, and facilitators noted by the RWG during the development of the reassessment process. Understanding these considerations, barriers, and facilitators will aid HTA agencies in designing and implementing reassessment processes.

Prior work on HTA reassessment emphasizes the value and benefit of meaningful stakeholder engagement—beginning early on and continuing throughout the reassessment process [20,21]. Because of the potential challenges involved in modifying longstanding clinical paradigms in response to recommendations, stakeholder resistance has been suggested as a main barrier to implementing a reassessment process [21]. There may also be limited support following the introduction of recommendations, which may result in misuse or disuse [22]. Building upon these lessons, the CanREValue Collaboration has made efforts to incorporate the perspectives of diverse stakeholders in the development of the framework.

Working collaboratively with and involving the HTA bodies in WG roles has helped to support the feasibility of adopting the RWG's recommendations to existing processes. As part of its update to the existing HTA processes, CADTH recently launched a consultation process [16]. Many of the recommendations by the RWG on reassessing cancer drugs using RWE align with CADTH's revised reassessment processes [16]. CADTH outlines four main triggers for reassessment: regulatory activity, reimbursement activity, questions about clinical and/or cost-effectiveness, or contextual changes [16]. Based on a clinical and economic review and consultation with key stakeholders including patient groups, clinicians, and drug programs, recommendations are made to reimburse, reimburse with conditions, or not reimburse [16]. Notably, CADTH's developed guidelines are applied to all drugs, with consideration of all available evidence, including and beyond RWE. Thus, the main emerging themes from the RWG may also be broadened to applications in non-cancer therapeutic context.

Globally, there is widespread interest in the implementation of processes enabling reassessments. In 2011, Brazil developed the National Committee for Health Technology Incorporation (CONITEC), with the aim of improving HTA and reassessment [23]. A recent analysis of their reassessment processes highlighted numerous opportunities for improvement, specifically surrounding the selection of candidates and methods of conducting reassessments [23]. Candidates for reassessment are often selected via the emergence of new evidence, increasing public interest, or the presence of inconsistencies amongst guidelines [23]. HTA agencies in other countries, such as the National Institute for

Health and Care Excellence (NICE) in the United Kingdom and the Institute of Medicine in the United States, have considered additional criteria when identifying candidates for reassessment [24,25]. The RWG's proposed guiding factors for selecting drug candidates for reassessments include unmet need and uncertainty in the magnitude of benefit of the drug. In the overall framework, the CanREValue Collaboration aims to proactively identify candidate drugs with high priority and policy-relevant uncertainties that can be addressed by RWE, rather than reacting to available evidence. As a result, the Planning and Drug Selection WG was established to address issues surrounding the identification and prioritization of uncertainties for reassessment [17].

As the landscape of oncology therapeutics evolves and funding decisions grow increasingly complex, there is a widespread need for drug reassessment following funding authorization. The CanREValue Collaboration established the RWG, which is dedicated to developing a comprehensive reassessment process. Continued collaborative efforts across the CanREValue Collaboration's WGs will independently but synergistically enable the timely development of a relevant, comprehensive, nationwide framework and will facilitate the path forward for the generation and use of RWE for the reassessment and refinement of funding decisions by decision makers.

**Supplementary Materials:** The following are available online at https://www.mdpi.com/artic le/10.3390/curroncol28050354/s1, Materials S1: Round 1–4 Survey Questions and Stakeholder Consultation: Questions for Consideration.

**Author Contributions:** W.F.D., E.C., B.F., J.A., D.B., N.B., P.C., C.C., B.J., E.L., G.M., T.P., A.R., T.R., J.M.B., A.D., R.E.M., A.P., M.T., P.T. and K.K.W.C. contributed to the conception and design of the contents of the framework. W.F.D. and VA co-wrote the manuscript. W.F.D., V.A., E.C., B.F., J.A., D.B., N.B., P.C., C.C., B.J., E.L., G.M., T.P., A.R., T.R., J.M.B., A.D., R.E.M., A.P., M.T., P.T. and K.K.W.C. critically revised the work and edited the manuscript, and approved the final manuscript. All authors have read and agreed to the published version of the manuscript.

**Acknowledgments:** The following are members of the CanREValue Collaboration Reassessment Working Group: E.C. (Co-chair), B.F. (Co-chair), Helen Anderson (BC Cancer), J.A. (Ontario Health (CCO)), D.B. (Saskatchewan Cancer Agency), Sylvie Bouchard (INESSS), M. Bryson Brown (Patient Representative), N.B. (PMPRB), P.C. (Government of Manitoba), C.C. (Alberta Health Services), Michele de Guise (INESSS), Marc Geirnaert (CancerCare Manitoba), Derek Finnerty (Patient Representative), Melissa Hunt (Health Canada), B.J. (Health Canada), E.L. (PMPRB), Helen Mai (CADTH), Suzanne McGurn (CADTH), G.M. (CAPCA), T.P. (PMPRB), A.R. (Dalhousie University), T.R. (CADTH), Daniel Sperber (pCPA), Maureen Trudeau (Sunnybrook Health Sciences Centre).

**Funding:** This work was supported by the Canadian Institutes of Health Research (Grant #HRC-154126). This study was supported by the Canadian Centre for Applied Research in Cancer Control (ARCC). ARCC is funded by the Canadian Cancer Society Grant #2020-706936.

**Institutional Review Board Statement:** Not applicable.

**Informed Consent Statement:** Not applicable.

**Data Availability Statement:** The data that support the findings of this study are available on request from the corresponding author.

**Conflicts of Interest:** D.B. participated in an advisory role with AstraZeneca. D.B. participated on an advisory board with Roche. E.C. owns stock in Abbvie US.

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
