# Peer review of "Considerations for Developing a Reassessment Process: Report from the Canadian Real-World Evidence for Value of Cancer Drugs (CanREValue) Collaboration’s Reassessment and Uptake Working Group"

_curroncol, doi:10.3390/curroncol28050354_

Round 1

Reviewer 1 Report

This paper is a report on the process and recommendations of one working group (RWG) of the high level pan-Canadian collaborative initiative (CanREValue) which aims to apply real-world evidence to reassess pan-Canadian health-care technology recommendations for reimbursement of selected cancer treatment technologies.  The importance of this topic has been recognized internationally, but there is as yet no recognized "gold standard" framework for implementing such a process on a national basis.  CanREValue is well positioned to design and implement such a process, and other developed countries may well draw on the Canadian model as they strive to create and implement their own mechanisms for utilizing real-world evidence in revisiting previous funding recommendations.

The composition of the RWG as described was appropriately diverse.  Industry was not included, but had the opportunity to contribute during the public consultation.  The framework development methodology was reasonable, and appropriately involved iterative interaction within the RWG, also between the RWG and other working groups of CanREValue and importantly with appropriate public consultation.

Given the appropriateness of the composition of the RWG and the rigour of its work process, its recommendations are ripe for dissemination.

Some portions of the manuscript require revision for clarification.  The distinctions between "reassessment" and "reassessment review"; also between "reassessment recommendation" and "reassessment review recommendation"  are not always clear.  

Examples:

lines 150, 151: "reassessment can be initiated..."  "Reassessment reviews may be identified.." Presumably "identified" implies flagging the need for reassessment at the time of initial HTA recommendation as described later.

Consideration 4 (line 219): "Barriers and Facilitators to the Uptake of Reassessment Recommendations", also Table 4.  The issues described seem to be barriers to the implementation of the recommended process of reassessment, i.e. following recommendation for reassessment, rather than the uptake of the recommendations from the reassessment as described in Consideration 3 "Reassessment Recommendation Categories", also Table 2.

Minor corrections suggested:

line 101: suggest "gaps are"

line 155: please clarify

line 171: "feasibly"

line 243: suggest insert reference

Reviewer 2 Report

It is great to see this Working Group engaging a wide range of stakeholders in an attempt to gain buy-in for the review of reimbursement decisions on the basis of real-world evidence. The results are not unexpected, and are not particularly original, however the value of this research is in the approach rather than conclusions.  This is made clear when the authors note in the discussion that the real aim here is to engage with stakeholders in order to increase the chances that the recommendations will be implemented.

The authors are successful in implementing their method, and the resulting article is valuable. However at some points throughout the manuscript clarity was lost and grammatical errors crept in. Below I highlight some examples:

Line 66 - "The generalizability of RWE is a major strength....". I don't think generalizability is the right word. In some ways RWE is less generalizable and more contextual. From the context it seems what the authors are trying to say is RWE is more relevant since it is the result of a natural experiment rather than a controlled experiment. 

Line 155 - "Public feedback on was collected..." can remove "on".

Line 123 - "Through an iterative, consultative processes, the...". "an" corresponds to the singular, so can't use it with "processes".

Line 132 - "Reassessment reviews may be identified by federal, provincial, or territorial (F/P/T) public drug programs when uncertainties are identified during the initial drug reimbursement review." Do the authors mean that the need for future reassessment is flagged if epistemic uncertainties arise during the initial drug reimbursement review?

Line 141 - "... the drug under consideration has replaced by another drug". Do the authors mean "has replaced another drug", or "has been replaced by another drug"? The two have entirely different meanings and it is not clear which.

Line 147 - "...guidance regarding evidence and data required for the..." In this context what is the difference between evidence and data?

Line 151 - "Further data collection and consideration for subsequent reassessment review should be prompted when there is high unmet need (i.e. no reasonable, publicly funded treatment alternatives available) or when there is substantial uncertainty in the magnitude of evidence despite clear signal of benefit and the uncertainty can be feasibility addressed by RWE study." This is a lot to take in. Can the sentence be broken up for clarity? What is meant by "magnitude" of evidence?

Line 170 - I think you can safely remove the sentence: "Reassessment review files may have different scope of review so there ought to be different types of review to accommodate differences in scope." I was left wondering what you mean by "scope", and I don't think it adds to the paragraph.

Line 189 - "...CADTH reimbursement review expert committee deliberative and recommendation frameworks...". That's a mouthful. Can this be simplified?

Line 190- "At the initial drug review, the evidence reviewed by experts includes considerations of patient needs, clinical data and opinion, cost-effectiveness, and ethical and implementation or adoption considerations". This list is a bit cumbersome. Suggest you separate out the clinical/effectiveness considerations from the ethical/implementation/adoption considerations. 

Line 211- "Similarly, facilitators to the implementation of reassessment recommendations included: generation of high-quality RWE, clearly defined reassessment criteria and recommendations, collaboration, and general facilitators" Remove "general facilitators" as it seems you are saying facilitators included facilitators, which is a tautology.

Line 279 - "HTA agencies in other countries, such as National Institute for Health and Care Excellence (NICE) in the United Kingdom and Institute of Medicine in the United States have consider additional criteria when identifying candidates 281 for reassessment [24,25]. The RWG proposed guiding factors for selecting drug candidates for reassessments include unmet need and uncertainty in the magnitude of benefit of the drug." "Consider" should be either "considered" or remove the "have". Also change "RWG" to the possessive form "RWG's"

Line 284 - "In the overall framework, the CanREValue Collaboration aims to proactively identify high priority policy-relevant RWE..." The last part of this sentence is a mouthful and doesn't make sense on inspection. The aim is to prioritise medicines for reassessment, not RWE perse. RWE is the tool not the aim.

In Tables 2 and 4 the phrase "clear and defined reassessment criteria" is used. Do the authors mean clearly defined?

Table 2, Recommendation 4, states "Facilitators to implementation included: generation of high-quality RWE, clear and defined reassessment criteria and recommendations, collaboration, and general facilitators." I suggest remove "general facilitators" as this is not specific enough, and change "clear and defined" to clearly defined.

Reviewer 3 Report

Wei Fang Dai et al. report on a preliminary Canadian framework for RWE-based reassessment of reimbursed cancer drugs, on behalf of the Re-assessment and Uptake Working Group (RWG) of the Canadian Real-world Evidence for Value in Cancer Drugs (CanREValue) Collaboration. The manuscript is well written and the presented qualitative research is important both from the scientific and societal perspectives, I recommend it for publication with a few minor revisions:

  • Both the Reassessment Working Group and the satellite external stakeholder consultation team involved multiple stakeholder groups (industry, patient, payer, etc) - however, all participant contributions are reported in an aggregated way, without detailed analysis of input by stakeholder groups and without data on the number and distribution of participants. It is suggested to report the number of active participants and survey respondents by stakeholder groups for all RWG meetings; and to report the 21 external stakeholder feedback messages by stakeholder groups. For example, the current text states that "some respondents suggested the removal of the recommendation category “do not continue funding/delist”... It would be interesting to see which stakeholder group(s) suggested the removal of the "delisting" recommendation category arguing that it could result in threatened medication access. 
  • Table 4, barriers and facilitators: the wording of barriers could be more clear in many cases. Examples: "Availability, reliability, and quality of RWE" is listed as a barrier to reassessment recommendations, although these are prerequisites; the barriers would be "unavailability, unreliability, or poor quality of RWE"; Another example: "Population size, and treatment volume" is listed as a barrier, although "low population size or treatment volume" would be the true barrier. Another example is "Willingness of decision makers to delist a drug, or increase funding relative to benefit": the barrier would be "Lack of willingness of decision makers...).   
  • Table 3, Description column, row 2: suggest replacing "effectiveness, safety, and cost-effectiveness of the initial review" to "effectiveness, safety, and cost-effectiveness of the investigated drug". 
